# Recent Applications of Multispectral Imaging in Seed Phenotyping and Quality Monitoring—An Overview

**DOI:** 10.3390/s19051090

**Published:** 2019-03-04

**Authors:** Gamal ElMasry, Nasser Mandour, Salim Al-Rejaie, Etienne Belin, David Rousseau

**Affiliations:** 1Department of Pharmacology & Toxicology, College of Pharmacy, King Saud University, Riyadh 11564, Saudi Arabia; rejaie@ksu.edu.sa; 2Faculty of Agriculture, Suez Canal University, Ring Road Km 4.5, Ismailia P.O. Box 41522, Egypt; ns_mandour@hotmail.com; 3INRA, UMR1345 Institut de Recherche en Horticulture et Semences, 42 rue Georges Morel CS 60057, F-49071 Beaucouzé CEDEX, Angers, France; etienne.belin@univ-angers.fr (E.B.); david.rousseau@univ-angers.fr (D.R.); 4Laboratoire Angevin de Recherche en Ingénierie des Systèmes (LARIS), Université d’Angers, 62 avenue Notre Dame du Lac, 49000 Angers, France

**Keywords:** seed, grain, multispectral imaging, hyperspectral imaging, quality evaluation, germination, viability

## Abstract

As a synergistic integration between spectroscopy and imaging technologies, spectral imaging modalities have been emerged to tackle quality evaluation dilemmas by proposing different designs with effective and practical applications in food and agriculture. With the advantage of acquiring spatio-spectral data across a wide range of the electromagnetic spectrum, the state-of-the-art multispectral imaging in tandem with different multivariate chemometric analysis scenarios has been successfully implemented not only for food quality and safety control purposes, but also in dealing with critical research challenges in seed science and technology. This paper will shed some light on the fundamental configuration of the systems and give a birds-eye view of all recent approaches in the acquisition, processing and reproduction of multispectral images for various applications in seed quality assessment and seed phenotyping issues. This review article continues from where earlier review papers stopped but it only focused on fully-operated multispectral imaging systems for quality assessment of different sorts of seeds. Thence, the review comprehensively highlights research attempts devoted to real implementations of only fully-operated multispectral imaging systems and does not consider those ones that just utilized some key wavelengths extracted from hyperspectral data analyses without building independent multispectral imaging systems. This makes this article the first attempt in briefing all published papers in multispectral imaging applications in seed phenotyping and quality monitoring by providing some examples and research results in characterizing physicochemical quality traits, predicting physiological parameters, detection of defect, pest infestation and seed health.

## 1. Introduction

The last couple of decades have witnessed growing concerns from governmental agencies to enhance seed quality parameters because without good seeds investments in fertilization programs, integrated pest management programs, water and other inputs won’t be worthwhile. Providing high quality seeds is very important for agriculture and seedling progression because seed quality is strongly related to the resistance to biotic and abiotic stresses, to the germination rate and to the plant performance [1]. Despite the methods used in crop planting such as manual transplanting of seedlings or broadcasting pre-germinated seeds, the quality of such seeds or seedlings is the key factor to guarantee high yields of the cultivated crops. The importance of seed quality is not a new issue, as all ancient nations recognized seeds as the main input for crop production and practiced seed screening as a means to improve crop yield and quality. In this regard, stakeholders involved in agricultural production such as farmers, traders, distributers and variety registration centers consider seed quality as an essential issue, not only to guarantee high production but also to secure more profit margins. Thus, national and international seed markets are always seeking the best descriptive characterization of seed quality traits. However, the seed quality term is an ambiguous and indefinite term because many definitions have been suggested based on different characteristics and attributes [2] and because the meaning of the term itself differs from the point of view of producers, farmers, manufacturers, and retailers. In general, the concept of seed quality is composed of several attributes, including varietal and analytical purity, germination capacity, vigor, viability, seed health and uniformity [3]. Seed health, germinability, physiological and physicochemical attributes of seeds, presence of weed seeds, presence of diseases and insect infestation are also included in the definition of the term “seed quality” [4]. In addition to the morphological and phenotypic features that are conventionally recorded by using visual inspection methods, the biochemical, genotypic and molecular markers are extremely important and ought to be also met. 

In most quality control programs, the evaluation of seed quality employing efficient and reliable methods for continuously checking fresh and stored seeds to explore their physicochemical properties, purity, germination, vigour, freedom from debris, dead or broken seeds and presence of pests, moulds or any other deterioration symptoms is an essential step. However, these continuous checking processes are not an easy task, particularly in the case of insects that are living and may be concealed inside the seeds with no external symptoms or signs of infestation. Suppliers tend to adopt reliable testing scenarios to provide trustful information about seed quality parameters. Some of this information is very easy to achieve, while other information related to viability, damage, vigor and health is very difficult to obtain directly due to time and technological constraints. Seed testing is generally carried out by detecting various physical, biochemical, physiological biomolecular markers which correlate well with the known seed constituents [5]. For improving seed quality analyses and to reducing the overall cost of labour-intensive tests, both International Seed Testing Association (ISTA) and Association of Official Seed Analysts (AOSA) recognize the importance of developing new technologies for expeditious, non-destructive seed quality determination [6]. Generally, any new method of seed quality evaluation should be globally accepted and thoroughly documented before being approved. 

Most of the methods employed in seed quality evaluation provide information about the overall quality of batches or bulks of seeds without referencing variation among single seeds that could be considered very important in plant breeding programs. For instance, a high vigor seed lot should hypothetically be homogeneous and produce a very narrow range of physicochemical and physiological properties, but under the effect of varied storage conditions, pathogen contamination or pest infestation, the overall quality of individual seeds within a seed lot will not be affected to the same extent, resulting in a wide spectrum of quality within this lot. Detecting the variations among seed lots and among individual seeds within a seed lot has great importance for both the seed industry and markets. Regrettably, the physiological and biochemical variation existing among individual seeds can’t be figured out from pooled seed samples, especially when some of seeds in the lots are dormant [7]. Actually, experimental procedures designed for exploring the quality of individual seeds are very few owing to a lack of specific techniques that consider single seeds as independent units having independent quality features. As the main aim of modern agricultural practices is not only to augment and enhance production, but also to produce safe and healthy food and feed of high quality, the removal of empty, diseased and infected individual seeds is substantially important. The visual inspection of seeds one by one by ordinary workers is very tedious and a time-consuming process and the performance of personnel working for continuous hours declines with time and it is sometimes less than satisfactory, which in turn results in inspection errors. In fact, development of on-site single seed screening seed units would be very useful not only for producers and farmers, but also for feed processors and food companies and other seed-related industries [8].

Recently, imaging techniques are gaining great attention in the seed industry for quality assessment and monitoring [9]. They were initially developed for measuring the morphological features of examined objects and have been widely used to visualize plants and plant parts including seeds [10,11]. Throughout the seed handling chain, sorting factors including purity and grading usually require visual inspection which in turn requires an objective tool to examine every single seed with minimum error [12]. In the framework of efficient implementation in the seed industry, imaging methods can play a strategic role for quality estimation and authentication. Besides their great ability in evaluating the overall quality parameters of seed lots, imaging-based techniques can be regarded as efficient tools to test and evaluate individual seeds to explore the imbibition process, to investigate the germination capacity and to discern vigor differences among seeds lots [5]. Nowadays, the role of imaging is growing in different agricultural applications due to the availability of cheap imaging devices, the increasing computational power of modern computers and the increased interest in non-destructive quality assessment strategies for seeds. As prerequisites for acquiring high-quality images, extreme consideration ought to be given for selecting the proper acquisition mode, illumination, detector type and relative position of seeds with respect to the camera [13]. As imaging techniques are characterized by their expeditiousness, accuracy and non-destructive nature, all stakeholders interested in seed quality are eagerly awaiting these innovative detection methods for improving the overall crop productivities.

### 1.1. Traditional and Advanced Methods of Seed Quality Evaluation

The ideal seed inspection protocols should include methods for seed measurements, field inspection, testing and retesting as well as seed certification. Seed testing is the process during which the essential characteristics of a seed lot in terms of varietal purity, moisture, germination, vigour are determined to enable the farming community to get high quality seeds. The ordinary practices of seed quality evaluation are usually performed by germination tests and can be followed by vigor tests and seedling growth characteristic measurement, especially because germination test might overestimate the seed potential [14]. The measurements of physiological and biochemical properties of a seed lot involves measurements of electrical conductivity, staining, and enzyme activity [15]. The traditional methods for evaluating seed quality are usually conducted destructively by using biochemical and molecular methods by experienced seed analysts with special chemicals and lab arrangements. Whilst proven to be efficient, such methods are criticized as being tedious, destructive, laborious and requiring trained seed analysts. For instance, an array of testing methods such as protein electrophoresis, gas chromatography (GC), high-performance liquid chromatography (HPLC), tetrazolium tests, leachate conductivity tests and accelerated ageing have been continuously developed and modified for characterizing seed vigor and to evaluate germination quality of seeds [16]. Although such methods exhibit high accuracy and good reliability, they have some limitations in terms of cost, time and labor requirements [4]. For example, the standard germination test is still practiced as the official method for testing seed viability and detecting abnormal, dead and dormant seeds. The percentage of dormant and germinating seed is required for labeling seed packages for sale [17]. 

For increasing the reliability and reproducibility, accurate and faster quality assessment methods with minimum human intervention are basically required to provide the best levels of seed quality for production and trade purposes [18,19]. Therefore, the potential of using non-destructive methods such as imaging techniques, NIR spectroscopy or precise remote sensors to overcome the limitations of conventional methods is gaining increasing attention [18]. The imaging techniques such as spectral imaging, thermal imaging, fluorescence imaging X-ray imaging, and magnetic resonance imaging offer reliable alternatives to the traditional destructive methods [20,21]. The X-ray imaging and magnetic resonance imaging are able to provide anatomical details and the spectral imaging, thermal imaging and fluorescence imaging are usually utilized to deliver the functional and nutritional information about the seeds being examined [21]. In general, the digital imaging approaches along with computer simulation are very useful for integrating complex data about seed quality parameters in an automated and objective manner. While colour imaging provides information about the morphological features (e.g., colour, size, shape and surface structure) of seeds being imaged using RGB (red, green and blue) cameras, seed radiography with X-ray imaging and nuclear magnetic resonance techniques have proved great potential in evaluating seed quality as well as efficient seed phenotyping in a non-destructive way by providing internal and anatomical features of the examined seeds. These techniques offer seed analysts and researchers detailed information about the seed maturity, internal structure, germination capacity, dormancy, viability, vigor, insect infestation and internal damages in embryo and endosperm [22]. However, some of these techniques are very infeasible in the case of large scales and may be harmful if wrongly implemented. Hence, these techniques will not be explained in this paper as more sophisticated details were discussed in some other sources (e.g., [21,23]). 

On the other hand, spectrometric technologies have been employed in determining various seed quality traits due to their practicality and ability to explore internal properties of biological materials. Depending on the frequency or the wavelengths used, the electromagnetic waves may be reflected, absorbed or transmitted through such materials, offering potential quantifications of many quality traits simultaneously [24]. This kind of non-destructive analysis depends basically on the resolution of the sensors and the mathematical chemometric model utilized. Classical spectroscopic techniques (fluorescence, visible, and infrared) are known to be generally quick and inexpensive to implement. They can be used in line as they do not require direct contact with the product to make a measurement. These methods can therefore be a remarkably effective alternative to traditional analyses that are generally expensive and technically long to implement. By the way, applying these spectral methods requires precise calibration not only for the tested seed cultivar but also for the trait(s) to be determined. Once calibration step is performed, a subsequent fast, nondestructive analysis of the seed as well as the simultaneous determination of multiple components can be easily achieved. Rahman and Cho [4] have reviewed some techniques including traditional color imaging, NIR spectroscopy and spectral imaging for the nondestructive assessment of seed quality traits. 

### 1.2. Spectroscopy and Spectral Imaging for Seed Quality Evaluation

Recently, optical sensing techniques, including spectroscopy and imaging systems, have been widely utilized for quality and safety evaluation of seeds. Systems based on spectroscopy provide spectral data either by spectroradiometry or spectrophotometry measurements. In spectrophotometry, the light reflected from a sample is related to the incident light as a measure of the spectral reflectance for the sample. In the case of spectrophotometers, a light source is also required, most commonly tungsten-halogen or xenon lamps [25]. In recent years, the near infrared spectroscopy (NIRS) method has demonstrated its ability to carry out simultaneous assessment of different quality traits with accuracy comparable to that of traditional wet chemistry analytical methods [26]. Depending on the sensor utilized, the spectrometer devices may cover the ultraviolet (UV) range (200–380 nm), the visible (VIS) range (380–750 nm), the shortwave near infrared (SWNIR) region (750–1700 nm) or the whole near infrared (NIR) range (780–2500 nm). The NIRS technology depends on the absorption of near infrared radiation by major constituents such as protein, lipids, carbohydrates and water resulting in a distinctive spectral pattern corresponding to the energy absorption and molecular vibrations relating to overtones and combination of C-H, N-H and O-H functional groups [27]. Scientific records of NIRS applications in seeds are sparse with much innovative implementation of this technique being in laboratory and industrial applications. Due to their advantages, near infrared spectroscopy (NIRS) techniques have been widely utilized for the quality analysis of crop seeds and kernels including quantitative analyses of oil content [28], protein [29,30], water [31], sugar, starch and carbohydrate content [8,32,33], viability and vigor detection [15,34,35,36], purity or variety discrimination [37], detection of disease and insect infestation [38], detection of physical damage [15,17] and other applications [4].

However, the spectral information extracted by this method is presumably limited to only a little portion of a sample where the measuring probe is positioned, without considering the spatial information. Thus, the sample being analyzed should be reasonably homogeneous in order to extract representative spectral fingerprint of the whole sample. This disadvantage of conventional spectroscopy can be easily alleviated by combining both spectral and spatial information by using hyperspectral (HIS) and/or multispectral (MIS) imaging techniques [39]. In addition to the spectral information that can be extracted from the images, spectral imaging systems are superior to traditional single-point NIR spectroscopy ones due to their capacity to show the spatial distribution of chemical constituents in the sample in forms called chemical images. Due its ability to combine the merits of spectroscopy and imaging, spectral imaging techniques have been invented to extensively investigate several aspects of seed quality determination [40]. Various applications in seed quality assessment using spectral imaging were reported, such as quality grading, disease and infestation detection, variety identification and classification, and prediction of chemical composition [16,41].

### 1.3. Importance of Developing Multispectral Imaging Systems

The unbridled desire of seed service centers to provide seed markets with high quality criteria seeds requires more concerted efforts and cooperative actions to develop advanced and innovative technological systems to provide unique quality that can satisfy all stakeholders’ expectations. The use of high speed digital image detection and computerized data processing to improve traditional seed quality analysis is increasing. The decreasing cost and increasing power of computer hardware and software packages for image processing have made image analysis systems more attractive in automatic inspection of different aspects of quality evaluation and sorting processes [39]. The improvement in image acquisition devices, together with new algorithms implementing expeditious image processing routines, has permitted quantifying many morphological features of seeds necessary for germination and vigour testing in a wide range of crop species [5].

Although conventional colour imaging systems can provide a number of essential quality traits for each individual seed within a seed lot such as area, perimeter, length and width, shape, surface colour, such a system is unable to provide detailed information about the chemical composition, structure and other invisible features of the seeds. On the other hand, spectral imaging systems in the visible to near infrared spectral range (400–2500 nm) have facilitated the development of imaging platforms having the capability for rapid quality assessment of grains and seeds based not only on the visual features of the seeds, but also on their intrinsic physicochemical characteristics. The correlation of these parameters with data obtained by traditional methods may also indicate the potential of these computerized imaging systems to generate new seed germination and vigour markers, suitable also for purity analysis and sorting [5]. Technically speaking, spectral imaging has been successfully applied in the classification of seed and grains, including maize, wheat, barley, soybean and rice and some other major food such as meat, fish, fruits and vegetables [42]. Owing to its ability to integrate both spectral and spatial information, this technique can be used to visualize local changes upon stress resulting from structural changes or from accumulation of specific compounds and their effect on cellular functions. Due to the great number of wavelengths and the time required of acquiring and processing spectral images as well as the redundant information among contiguous bands, the spectral imaging system is not suitable to be implemented as it is for on-line applications. Generally, the ultimate aim of using hyperspectral imaging systems is to establish a multispectral imaging system as an essential part of a computer-integrated machine vision system for different on-line applications [43]. Due to the complex nature of samples, there is no single magic algorithm valid for all image analyses applied to hyperspectral images to select the most prominent wavelengths and to bridge the semantic gap between spectral data representations and efficient image visualization, but it rather needs deep experience from the developers.

Building an integrated computer-aided image analysis system for overall quality evaluation processes of seeds could be realized by combining colour imaging and spectral imaging in one system (Figure 1). The colour imaging system will be used to provide measurements of the morphological features (dimensions, colour, shape, texture … etc.), germination capacity (radicle elongation, timing of germination, germination speed, vigor etc.) and seed health conditions (diseased parts, pest infestation, nutrient deficiency … etc.). Along with the colour imaging system, the spectral imaging system will be utilize to provide spectral information about the examined seeds to give detailed information about the chemical compositions (protein, lipid, moisture, pigments etc.) of the examined seeds. Installing both systems in one unit will augment the potential of such systems in solving complicated problems and dealing with challenges that would not have been tackled with either system alone. Data modeling algorithms and pattern recognition tools should be also integrated to develop robust models to visualize the hidden information in both kinds of images and then installed in a master computing system to supervise the whole procedure of quality inspection process.

## 2. Colour, Hyperspectral & Multispectral Imaging

Similar to the human eye, traditional colour imaging known also as RGB imaging uses three broadband colour channels (Red, Green and Blue) to produce a single color value for each pixel in the image [18]. In the traditional mono CCD cameras, the sensor is coded with a mosaic arrangement of pigments that transmit red, green and blue. These pigments make up what is called colour filter array (CFA) with broad, overlapping transmission characteristics. The CFA is manufactured in a way that one quarter of the sensor pixels sees red, one quarter sees blue and one half sees green. Therefore, the colour image could be thought as three separate greyscale images combined together to form a colour image. As a consequence of the broad wavelength band ranges recorded by RGB cameras, RGB imaging has very limited spectral resolution and a large amount of information is lost, which makes RGB imaging unsuitable in differentiating similar samples that only show separate spectral variations within a single broadband range. This phenomenon is called metamerism which means that two different objects having different spectra may appear identical to the eye with the same color under a given illumination and may look different when they are viewed under another illumination. For example, both chlorophyll *a* and chlorophyll *b* simply look green in RGB images although they are distinctly different in terms of their spectral patterns. Metameric colours (metamers) represent a severe problem in many applications. For instance, in seed health diagnosis, the traditional trichromatic camera or the human eye are unable to discern the healthy and diseased tissues when spectral variations between these two classes are small and the appearance of both tissues looks the same under certain illuminating conditions. There have been numerous research attempts to utilize color as the only parameter for distinguishing seeds of different quality grades [44,45]. Based on these facts, colour alone did not adequately describe all differences among seeds of different quality grades. Although image analysis of RGB images can provide useful information about the overall quality of grains and seeds, information related to chemical composition can’t be extracted from such RGB images because they are limited to the visible range of the spectrum. Thus, to resolve such problems resulting from metameric colors and limitations of providing chemical-related information, images of the seeds being examined are preferably acquired with a spectral system not only to extend the number of imaging channels beyond the conventional three channels but also to measure the reflected light at different wavelengths for every single pixel in the image which is very beneficial for a wide range of applications.

Instead of using three broad bands of colour for providing a stack of only three separate greyscale images showing the overall red, green and blue light reflectance, multispectral imaging provides calibrated reflectance data at multiple ‘*discrete*’ wavelength bands spaced over an extended spectral range from ultraviolet (UV) to near infrared (NIR) region (Figure 2). The resulting multispectral image is a stack of many greyscale sub-images showing the exact relative light reflectance at many non-overlapping ‘*discrete*’ wavelength bands along a range beyond the human visual perception. Therefore, the technique of acquiring a multispectral image is usually called multi-channel imaging because it can capture image data at specific multi-wavelengths across the electromagnetic spectrum which provide the information required for the characterization and identification of the components composing the seeds being analyzed [40]. Besides concurrently supplying the spatial and spectral details of the target seeds, multispectral imaging does not require prior sample pre-treatments for simultaneous assessment of multiple quality traits. 

When spectral images are acquired for very narrow spectral bands rather than discrete bands, the system is called hyperspectral imaging. The main difference between multispectral imaging and hyperspectral imaging techniques is number of wavebands at which the spectral image is acquired. In a hyperspectral imaging scenario, the spectral image is recorded by utilizing a great number of wavebands leading to a continuous spectral range. On the contrary, the spectral image is recorded for a few discrete bands in the case of multispectral imaging [46]. Thus, hyperspectral images can provide a full continuous spectrum for every single pixel in the image (Figure 2j). In regard to the number of spectral information provided by each system, colour imaging provides information at only three channels (R, G and B) within the visible range, multispectral imaging provide information at few discrete channels (up to 20 wavelength bands) located on different regions of the electromagnetic spectrum [47] and the hyperspectral imaging facilitates spectral information at so many wavelength bands not only in the visible range but also extended outside the visible ranges of the electromagnetic spectrum [48]. As the multispectral imaging system is constructed using fewer spectral channels, this implies shorter acquisition and processing times than the hyperspectral imaging system. A true hyperspectral image consists of hundreds of neighboring wavebands for each spatial position of a sample with every pixel contains the spectrum of that portion. Thence, hyperspectral imaging system could be considered as a potential alternative to the point-based spectroscopic techniques used in a wide range of applications in the agro-food sectors [49] and the sensing and visual capabilities of this technique have so far been exploited in numerous applications for seed quality assessment such as variety discrimination [50], defect detection [51,52], prediction of chemical composition [53] and determination of viability and germination capacity [41,54,55] in different sorts of seeds including maize, wheat, barley, oat, soybean, and rice seeds. By the way, the full wavelength spectrum of hyperspectral data can be reduced to several key wavelengths that can be implemented to develop a multispectral imaging system with enormous advantages [56]. Using fewer wavebands can lead to faster sensor systems and shorter acquisition time. 

### 2.1. Image Acquisition Methods

Due to the fact that all spectral imaging systems (either hyperspectral or multispectral systems) are required to gather information from the target objects in the spatial and spectral domains, it is very important to supply such systems with an acquisition mechanism to scan the sample in both directions simultaneously and to make the light sensor records light intensity at every pixels at every wavelengths. 

Based on the applications and the available optical devices, there are three methods of acquiring spectral imaging data sequentially, namely point-scanning (whiskbroom scanning), line-scanning (push-broom scanning) and wavelength-scanning (area or plane scanning) methods. The first acquisition method depends on acquiring the data point by point which required to systematically scanning the target seed samples in both X and Y directions as shown in Figure 2a. In point detector spectrometer, a full spectrum is acquired for each pixel in the scene. Thence, at the end of each step, a complete spectrum of only one point (pixel) was recorded at a time (Figure 2d) and the acquired data are stored as band-interleaved-by-pixel (BIP) format. This design is commonly used for microscopic imaging where the acquisition time is usually not a problem [57]. Examples of point scanning systems are: atomic force microscopes, confocal microscopes, and X-ray photoelectron spectroscopy. Instead of scanning the samples with a point-wise manner, the second method for image acquisition (Figure 2b) records the data line by line by moving the target sample systematically in X direction only. After passing through an entrance slit, the light reflected from the line is dispersed by a two-dimensional dispersing element (spectrograph) and then recorded on a two-dimensional detector array in which one spatial dimension was used to store the spatial details of the scanned line and one spectral dimension was used to record the dispersed light from the line at constituent wavelengths. In each movement in the X-direction, a 2-D image (y, λ) having one spectral dimension (λ) and one spatial dimension (y) is recorded. By scanning all lines of the target sample in the X-direction, full image data were created and then saved in a band-interleaved- by-line (BIL) format. This configuration is believed to be suitable for industrial applications where the examined objects are scanned during their movement on a conveyor under the imaging system. In general, acquiring full spectrum for every single point of the target sample at very narrow bands with hundreds of wavelengths makes the previous two methods of acquisition unsuitable for acquiring multispectral images if shorter acquisition time is targeted. Also, one disadvantage of these types is that its performance strongly depends on the alignment of the optical elements; therefore, the calibration is crucial and can be very costly. In the case of area or plane scanning method as shown in Figure 2c, one spectral image is recorded by sequentially scanning the whole spatial scene of the target object one wavelength after another without any movement for either the imaging unit or the target object and the resulting imaging data was stored in a band sequential (BSQ) format. To scan the scene with at a certain wavelength, this method of image acquisition requires a rotating filter wheel (Figure 2c) or a tunable filter such as liquid crystal tunable filter (LCTF) or an acousto-optic tunable filter (AOTF) for sequential changing the wavelength at which the image is acquired. In general, there are two possibilities in locating the filters: either in front of the camera or at the broadband light source to illuminates the scene with only light of the required wavelength. For capturing an image at a new wavelength, the filter mechanism must be operated to the desired filter which makes this type of acquisition suits unmoving targets. 

Generally, at the end of ‘one scan’, the point scanning method provides a full spectrum for only ‘one point’ of the target sample (Figure 2d) and the line scanning methods provides a full spectrum for only ‘one line’ of the target sample (Figure 2e); meanwhile the area scanning approach provides a full spatial view of the entire object at only ‘one wavelength’ (Figure 2f). Despite the modes of acquisition, at the end of the whole scanning processes for the entire target sample, a three-dimensional block of data called ‘*data cube*’ or ‘*spectral cube*’ $R_{ijk}$ is formed with two spatial dimensions (*x, y*) and one spectral dimension (λ) as shown in Figure 2g. The *i* index corresponds to the first spatial direction, the *j* index corresponds to the second spatial dimension, and the *k* index represents the spectral dimension. For example, a scene with 1024 × 1024 spatial samples and 20 wavelength bands in the visible range (380–780 nm) will constitute a 1024 × 1024 × 20 data cube, meanwhile an RGB image of the same scene is a 1024 × 1024 × 3 data cube revealing that a great deal the spectral information of the scene being images is lost in case of the later one.

The acquired spectral image has plethora of data and the formed data cube could be viewed either in the spatial or the spectral domains [18]. As shown in Figure 2h, a data cube could be viewed as separate 2D greyscale sub-images (slices) showing the same target object at different wavelength bands. Similar to the usual RGB image which composed of a cube with only three greyscale slices (i.e., R, G and B colour channels) in the wavelength dimension, the spectral image is composed of many greyscale slices. The number of these greyscale slices is equivalent to the number of wavelength bands used during image acquisition. As a mean of simplicity, any three of these grey sub-images (slices) could be easily concatenated one after another (one slice for each color channel) to form a pseudo-color image for the target sample as shown in Figure 2i. On the other hand, the same data cube shown in Figure 2g could be also viewed in the spectral domain by viewing the spectra of any pixels in the spectral image. If plenty of narrow contiguous wavelength bands were used in building up the data cube (i.e., a hyperspectral image) one full spectrum for every pixel in the image could be viewed as continuous solid line as shown in Figure 2j. By using a few discrete and uncontiguous wavelength bands in forming the data cube (i.e., a multispectral image), the values of light intensity at every pixel could be plotted against these discrete wavelength bands as represented by the histograms shown in Figure 2j. For the demonstrative example shown in Figure 2g, there are possibly seven non-overlapping greyscale sub-images (slices) captured in the spectral dimension yielding a multispectral data with only seven columns representing the grey level intensity of the target sample at each discrete wavelength band (Figure 2j). The figure also demonstrates how the appearance of the target sample changes at these wavelengths from very dark (low intensity) at 450 nm to very light (high intensity) at 850 nm.

### 2.2. Illumination-Based Multispectral Imaging Systems

Based on the relative position of the light source to the detector (lens, spectrograph and the camera), spectral images could be acquired in reflectance, transmittance or interactance modes based on the way by which the light beam is received by the detector after interaction with the examined sample [58]. Reflectance configuration, in which the light source and camera are in the same direction, is very common in both hyperspectral and multispectral imaging on the grounds that the reflectance values can be simply converted to absorbance values using the expression (Absorbance = log(1/Reflectance)) to fit the Beer-Lambert law [59]. Compared with hyperspectral imaging systems, multispectral imaging systems can be more easily designed in various configurations based on the structure of the spectral device and the illumination systems used during image acquisition. For instance, spectral devices could employ similar principles of colour imaging for generating multispectral images by patterning the sensor with micro-sized optical filters called multispectral filter array (MFA) that incorporates proprietary pixelated band-pass filters whose transmission characteristics can be precisely tuned during manufacturing [60]. Thus, the raw spectral image captured with such multispectral cameras consists of separate greyscale sub-images each taken at a specific wavelength with a predefined bandwidth. Accordingly, different multispectral device designs could be manufactured with custom sets of bands either in the visible or invisible regions of the electromagnetic spectrum based on the applications required for detecting various quality traits. Also, multispectral cameras can filter light reaching the surface of the image detector by utilizing specific band-pass filters to permit recording concurrent images at various wavebands. This conventional configuration of multispectral imaging system modulates the incoming light based on the different transmission spectra of the filters. This makes the decoupling of the reflectance properties of the material and the incident light very difficult in addition to position mismatch, changes in imaging focal length and uneven brightness [61]. Although these filter-based systems are greatly adaptable, they are inherently slow and costly [62]. 

In fact, the lighting is considered as significant as the optical parts, because it has the essential role in conveying the primary information of the inspected specimens. Hence, changing light geometry may lead to different images even when using the same camera. A well-designed illumination system helps in improving the accuracy of the imaging system, increasing contrast in the acquired scenes, reducing the time and complexity of the image analysis steps, leading to successful image analysis, and decreasing the cost of the image processing routines [63]. Although a broad flat spectrum illumination with an equal intensity (e.g., tungsten-halogen lamps) is required by a hyperspectral camera to acquire hundreds of sample frequencies across the spectrum, multispectral images may be acquired under different illumination conditions at different ‘*discrete*’ wavelength bands. Instead of using filter wheel or tunable filters, the approach of using various illuminants of different spectral power distribution depends on ‘*sequentially*’ illuminating the scene with different narrow-band wavelength light sources, which can reduce or eliminate light scattering caused by non-flat or irregular surfaces. As each discrete wavelength light source illuminates the sample, a 2D greyscale image is acquired using a camera with a uniform wavelength response. Because the number of wavelengths is limited, this approach is only suitable for acquiring multispectral images. In this approach, instead of using tungsten- halogen lamps, multispectral images could be acquired by illuminating the sample with narrow-wavelength-range light emitting diodes (LEDs) of known spectra. Contrary to the single broadband light source of fixed emission characteristics (like a halogen bulb), using strobed LED illumination at multiple specific wavelengths allows the illumination conditions for each different sample type to be optimized. By taking greyscale images under the illumination of these LEDs in turn, each at a successively longer wavelength, one can build up a set of registered images that contain more information than a simple RGB image under white-light illumination. During operation, the object is lit with a set of distinct LED illuminations and a synchronized monochromatic or RGB camera captures the corresponding images. Most multispectral LED imaging systems, with a few exceptions, often use monochromatic cameras owing to its high sensitivity compared to the colour camera. By activating the illuminations one by one and concatenating the acquired imaging synchronously, a multispectral image of the scene is obtained. As the cost of such a system is presumably low; producing commercial LEDs at different wavelengths throughout the electromagnetic spectrum has been gradually increased over the last 20 years. In a very short time, LED lighting has widespread in agricultural, industrial and household settings, which opens new venues of real applications of high-power narrow-waveband LED devices. In fact, it is easier to create an illumination unit having fast changing spectra than a camera having changing spectral sensitivity. Thus, compared with the optical filter configuration that requires longer acquisition time for changing the filters in the filter wheel or in the tunable filter device, multispectral imaging design supported with different LED illuminants requires shorter acquisition times and best suited for moving objects [64].

From mathematics point of view, a multispectral signal at any point in a scene *I_mn_* can be modeled as a radiometric transfer process measured at a pixel (*x*,*y*) in the *m*^th^ channel and *n*^th^ illumination and is given by the following equation:(1)$$I_{mn}=\int S\left(\lambda \right)c_{m}\left(\lambda \right)p_{n}\left(\lambda \right)d\lambda$$
where *S*(*λ*) is the spectral reflectance of the scene point, *c_m_*(*λ*) is the spectral response of the camera at *m*^th^ color channel, and *p_n_*(*λ*) is the spectral power distribution of the *n*^th^ illumination. If a monochromatic camera was used with only one channel (*m* = 1), a multispectral image with *n* sub-images would be obtained and when an RGB camera was used with three color channels (*m* = 3) and *n* illuminations, then by stacking the *n* images together, a multiplexed multispectral image with 3 × *n* sub-images would be obtained. The spectral reflectance *S*(*λ*) of a point in the scene is determined by the imaged material itself that characterizes the property of the object. If the scene’s spectral reflectance is given, it is possible to calculate the imaging result under different combination of any type of camera and light source [61]. Therefore, to add more information in the acquired multispectral images, the target samples could be imaged in the visible and invisible ranges of the electromagnetic spectrum to increase the discriminating capacity of the system for better characterization of the biological objects being imaged [12]. More interestingly, when the objects are illuminated with LEDs having peak emission in the blue or UV region of the spectrum and observe the spectra emitting in the green or red regions, fluorescence signals from the objects being analyzed can be captured which adds new venues of useful applications of this techniques in detecting various *fluorophores* (i.e., chemical constituents that are able to produce fluorescence emission up on light excitation) such as chlorophyll or fluorescent pigments in seed samples [65,66].

Figure 3 shows a schematic representation of the key components of a multi-LED multispectral imaging system configuration. This demonstrative system is supported with an array of 12 LEDs of different nominal emission wavelengths. The LEDs are usually fixed on a printed circuit board (PCB) and their emission intensity is automatically controlled by a computer through an USB interface [67]. Each LED is sequentially switched on for a definite period during image acquisition and synchronized with the camera trigger using a pre-programmed microcontroller. As a result of turning on the LEDs one after another, a total of 12 greyscale images were acquired by the monochromatic camera (as shown in the computer screen in Figure 3) which combined altogether to form one multispectral image. If an RGB camera was used instead of the monochromatic camera, one colour image (with 3 channels each) would be acquired when turning on each LED leading to a multispectral image of 3 × 12 = 36 greyscale images. Generally, the optical devices and LED illumination is usually placed inside a dark enclosure to prevent any light interference from ambient lights. Similar designs with the same principle are currently manufactured by VideometerLab Company (Videometer A/S, Hørsholm, Denmark) and have been used intensively for different applications not only in seed science and technology but also applied for quality studies of many products [18,40,68,69,70,71,72]. Even though it is supplemented with special software package for acquiring the processing the images of the samples, this system is still not totally automated [18]. In this design, the exact strobe time of each LED could be easily optimized for each object type using built-in software’s light setup wizard that can be saved for all subsequent image acquisition processes for similar objects.

In recent years, great attention has been paid to this type of LED illumination-based multispectral imaging system due to its robustness, rapid controlled switching ability and cost effectiveness [73,74]. The accessibility to different colour and high intensity LEDs of peak wavelengths in the ultraviolet, visible and infrared regions facilitates the construction of cost effective illumination-based multispectral imaging systems. To ensure the highest possible reproducibility of images and to guarantee reliable extracted information, the whole system especially the camera and illumination units should be appropriately calibrated before acquiring the images by adjusting intensity of the reflected light to avoid sensor saturation. Like all modern imaging techniques, even minor variations in the exact setup of light source, physical geometry, sensor sensitivity and imaging method can affect the data quality and hence the ability to apply robust models that give the required information to evaluate the target samples [12]. Moreover, the overall calibration procedure is very crucial since any variations in environmental conditions, such as temperature and/or relative humidity, can cause the camera to lose calibration. 

## 3. Chemometrics and Spectral Analysis

The acquired multispectral images are very abundant in information in the spatial and spectral domains, so that bias-free methods for data mining from the acquired images are very crucial, especially if one desires to analysis several components simultaneously [18]. These multispectral images provide a detailed record of the spectral variation over heterogeneous samples, to which statistical methods could be applied to reveal differences among and within samples and to obtain an overview of the underlying patterns that would not be ‘*seen*’ otherwise. Analyzing complex spectra having relatively weak or overlapping bands poses a great challenge [41]. When many combinations of variables are involved such as different wavelengths and different seed quality traits, the generated data are immense which implies some challenges in thoroughly interpreting and analyzing the data. In such situations, *chemometric* data analysis can underpin the dominant patterns in large data matrices in a fast and robust manner. The term ‘*chemometrics*’ is originally applied in chemistry, where huge data matrices are generated by developing mathematical and statistical methods to extract useful information related to the chemical characteristics of the analyzed seeds. Although the multispectral images contains very useful information about absorption bands related to overtone and combination vibrations of C-H, N-H and O-H functional groups of the major chemical constituents of the tested seeds, dealing with all of this information is a challenge itself [27]. Data extracted from images can be used to describe differences among samples (e.g., variety discrimination and/or seed health detection) or to predict quality parameters such as chemical combustion (proteins, lipids, moisture, etc.). The transition from images to data models helps in reducing the redundant information involved in the images to other forms of information relevant for specific applications. 

Based on the purpose of analyses, there are many calibration algorithms and pattern recognition methods that are widely known to perform well for qualitative (classification) and quantitative (prediction/regression) analyses of seed quality. The classification modeling method could be also classified into supervised or unsupervised methods. The supervised techniques are used to build classification models to classify new unknown samples to predefined known classes on the basis of their spectral signatures; meanwhile, the unsupervised methods do not require a prior knowledge about the grouping of the samples [57]. Despite the methods used for classification or prediction scenarios, the developed models for multispectral data could be linear such as partial least squares regression ‘PLSR’ [15,75], partial least squares-discriminant analysis ‘PLS-DA’ [17,36,76], normalized canonical discriminant analysis ‘nCDA’ [77], extended canonical variates analysis ‘ECVA’ [2,35,78]; or nonlinear such as support vector machine ‘SVM’ [15,54,79,80], k-nearest neighbors algorithm ‘k-NN’ [17,81] and back-propagation neural network ‘BPNN’ [76]. Once a calibration model is developed, it must be properly validated by using the model in predicting quality traits in independent seed samples, not related with samples included during calibration step [17]. However, due to light scattering, baseline shift, instrumental drift or path length differences, the spectral data from seeds may exhibit unwanted systematic variations that are not associated with their intrinsic chemical composition. Hence, before extracting any information related to the seed characteristics or developing the relevant chemometric models form their corresponding spectral data, some mathematical pre-processing treatments could be applied to eliminate or minimize such systematic noise from the original spectra and to increase the signal-to-noise ratio. These pretreatment mathematical approaches involve mean centering, auto-scaling, normalization, derivatives, multiplicative scatter correction (MSC), standard normal variate (SNV), and other techniques [41,47]. In general, spectral bands can be evaluated explanatively, and if there is a pattern between absorption data and reflectance measurements, the next step is to use a classification or prediction model. From the spatial domain aspects, different image processing routines such as edge detection, filtering, band ratio, thresholding, skeletonization and textural analysis algorithms could be implemented individually or in parallel to extract other extra informative features about seed inherent quality [39]. Above all, despite the modeling algorithm used or spectral preprocessing methods applies, it is very important to have appropriate sampling protocols, accurate reference analysis methods, reasonable spectral precision and excellent presentation of the samples to the imaging instrument [59].

## 4. Applications of Multispectral Imaging for Seed Quality Monitoring

Because the images collected by the multispectral imaging system are in a three-dimensional form, they could be used to analyze minor and subtle variations in the physicochemical properties of tested seeds. This technology has shown promising results in determining different seed quality features. By the way, applying this technology in seed quality evaluation is based basically on the rigorous understanding of the technology fundamentals and how to link seed quality parameters with the information available in the images [63]. Because the multispectral imaging system has gained widespread acceptance for agro-food, remote sensing, medical and pharmaceutical applications, it has been used in different studies in quality evaluation of different products, for example to predict physicochemical parameters in blue-veined cheeses [82], quality assessment of pomegranate fruit [83], classification of onion bulbs [56], freshness evaluation of rocket [84,85], quality monitoring of pharmaceutical tablets [86], skin characterization and dermatology [87], characterizing sensory properties and physicochemical parameters in meat [88,89], beef microbial safety identification [40] and adulteration detection in beef [47,68,90].

Although a lot of articles reviewing the potential usages of hyperspectral imaging, RGB imaging, X-ray imaging systems for seed sorting and testing are available [5,13,16], there is only one recent literature review paper prepared by Boelt et al. [6] just discussing the possibility of using multispectral imaging in detecting fungal infection in barely seeds and maturity development of sugar beet and spinach seeds. Therefore, a much more comprehensive review is needed to highlight recent research endeavors in using fully-operated multispectral imaging systems in seed phenotyping and quality monitoring of different sorts of seeds by providing some examples and research results in characterizing physicochemical quality traits, predicting physiological parameters, detection of defect, pest infestation and seed health. The feature-related wavelength bands can be straightforwardly extracted from the whole spectrum by analyzing hyperspectral imaging data using some well-known wavelength selection algorithms [91,92] and these selected wavelengths represent the suitable candidate bands for building an independent multispectral imaging system for the features in concern. In this regard, independent multispectral imaging systems with different designs have been used successfully in different sort of applications in detection and identifying numerous quality features of different seed cultivars. This review article highlights only those research attempts devoted to implementations of fully-functional multispectral imaging systems and does not consider those research attempts that just selected some key wavelengths extracted from hyperspectral data analyses without building fully-functional multispectral imaging systems. Moreover, applications presented in this article only review the research works related to quality evaluation of individual harvested or stored seeds. Although monitoring procedures of seeds during their growth in the open field before harvesting are extremely important because they can provide valuable information about seed overall quality and detailed records about any possible infection or disease invasion, the applications of aerial multispectral imaging systems in the open fields will not be discussed here and should be sought elsewhere (e.g., Dammer et al. [93]). All of these factors make this article the first attempt at summarizing all published multispectral imaging applications in seed phenotyping and quality monitoring as concisely detailed in the following sections and as listed in Table 1 below.

### 4.1. Seed Purity and Variety Discrimination

Discrimination among seed varieties and checking the seed purity plays a key great role in plant breeding programs, seed production, genebank management and in the general trading of seeds. Testing seed purity involves checking the presence of plant debris, foreign materials, weed seeds, contaminating species and broken and damaged seeds. Separating varieties and determining their critical properties in terms of distinctness, uniformity and stability standards are extremely significant for variety registration, intellectual property rights of plant breeders, as well as for developing new varieties in the markets [94]. Measurements of some essential morphological features are very vital in varietal identification and cultivars discrimination to highlight the phenotypic variations, but in some occasions these features may be not enough to act as discriminators among such cultivars. For instance, molecular markers and other biochemical methods are usually used for the identification and characterization of germplasm [79]. The traditional method currently used in international genebanks is usually done by visual inspection of a huge number of seed varieties to evaluate their phenotype features and to determine whether such varieties are completely new or have already been registered in the genebank. As these procedures are very time consuming and require highly skilled seed analysts, the development of a non-invasive, rapid and reliable technique for identifying and distinguishing the purity of verities is of boundless advantages [6]. 

Recent studies have demonstrated the ability of multispectral imaging in tandem with multivariate analyses for evaluating seed purity and varietal discrimination. The potential of using this technology in identifying and characterizing seed variety on the basis of differences in spectral signature and the ordinary morphological features was investigated in [65,76,77,81,94].

By using illumination-based multispectral imaging system in the visible and near infrared region (375–970 nm) for varietal identification and discrimination tomato seeds, Shrestha et al. [94] examined eleven varieties either all together or in a pairwise manner with their offspring using their spectral fingerprints by the aid of multivariate analysis of normalized canonical discriminant analysis (nCDA), principal component analysis (PCA) and partial least square discriminant analysis (PLS-DA). The nCDA model was used to minimize the distance within classes and to maximize the distance between classes. The developed models displayed overall classification accuracy of 100%, 85% and 82%, respectively. In reality, the nCDA method was employed as a supervised transformation building method to divide the image into regions of interest of different spectral signatures. When two different varieties have different spectral features, they will be appeared as two different region of interest in nCDA transformed image which will be easy to interpret and simple to segment for subsequent data analysis. When Olesen et al. [79] tried support vector machine discriminant analysis (SVM-DA) in discriminating tomato seeds of different cultivars, they achieved 100% discrimination accuracy. The same Shrestha et al. research group [95] used the same methodology in discriminating among tomato seeds of different maturity stages by following the same normalized canonical discriminant analysis (nCDA) algorithm. The data analysis showed a higher classification accuracy of tomato seed maturity measured by their germination potential. The result also suggests that seed maturity determination should be based on seeds instead of judging by fruit maturity or colour.

With the same technique, Hansen et al. [65] collected both reflectance and fluorescence emission spectra by installing low-bass filters at 500, 600 and 700 nm in front of the CCD camera to discriminate twenty diverse varieties of rice. In total, each pixel contains 46 spectral values representing all possible chemical characteristics of the examined seeds. After segmenting multispectral imaging to locate rice seeds in a clear background, a set of 177 morphological and spectral features extracted from each single seed was modeled using canonical discriminant analysis (CDA) projections and k-nearest neighbor (k-NN) classifier to discriminate and enhance differences among rice seed varieties The results revealed that the proposed method was able to identify potential off-types within the seed lots of each accession with an accuracy of 93%., which may indeed be useful in managing genebank accessions by including only new accession and avoiding maintaining seeds that does not represent novel genetic diversity. 

For differentiation of wheat varieties and triticale varieties, Vrešak et al. [81] used a VideometerLab multispectral imaging system with 19 bands (375–950 nm) for acquiring multispectral images of seed samples and then extracted both morphological features (e.g., color, texture and major dimensions) as well as features from nCDA transformation images. Based on all extracted features, a *k*-nearest neighbor (*k*-NN) based model was built to discriminate different wheat varieties. The overall results on varieties separation using the proposed method revealed that Agostino variety was classified with accuracy of 97.4%and the overall classification accuracy for the rest of varieties was below 67%. They reported that, discrimination accuracy could be greatly improved up to 95.8% in case of grouping varieties together according to visual similarity.

The feasibility of using LED illumination-based multispectral imaging (405–970 nm) associated with chemometric data analysis was examined by Liu et al. [76] for nondestructive discrimination of transgenic and non-transgenic rice seeds. Different chemometric data analysis methods such as principal component analysis (PCA), partial least squares discriminant analysis (PLS-DA), least squares-support vector machines (LS-SVM), and PCA-back propagation neural network (PCA-BPNN) were tested to see their performance in classifying rice seeds according to their genetic origins. The results revealed that LS-SVM model method exhibited the highest classification accuracy (100%) in distinguishing insect-resistant *Bacillus thuringiensis* transgenic rice seeds from non-transgenic ones. In another study, the same authors [96] followed the same routines in discrimination between conventional and glyphosate-resistant transgenic soybean seeds and their hybrid descendants by using the same multispectral system. Similarly, the results from this study attested the capability of the multispectral imaging supported by BPNN model in differentiation among conventional and glyphosate-resistant soybean seeds and their hybrid descendants with an excellent classification of 98%. In both studies, combining spectral data of the seeds with their corresponding morphological features extracted from their multispectral images (*L*, a*, b** colour values, area, length, width and roundness) produced better predictive accuracies and helped in improving the discrimination for all examined multivariate analysis algorithms.

Because the sorting process of seeds based on certain genetic features is a very time-consuming process that can take thousands of hours of labour to complete for a single average-sized breeding programmer, De La Fuente et al. [77] evaluated the ability of the VideometerLab3 multispectral imaging system to discriminate haploid from diploid maize seeds, testing the system on several maize genotypes that display varying difficulty in manual sorting. In addition to the 19 illuminated wavelength bands used for recording reflectance images, four band-pass filters at 400, 500, 600 and 700 nm were used for measuring fluorescence from the kernel surface producing a total of additional 60 excitation/emission combinations of fluorescence. The canonical discriminant analysis (CDA) method was used for segmenting the kernels form the background and to extract the main reflectance and fluorescence data either from the whole kernel or from embryo portions. The system has shown a great ability in discrimination between hybrids and haploids with accuracies ranging from 40% to 100% for different six genotypes. 

While hyperspectral imaging systems with relevant multivariate analyses are able to identify kernel damage, multispectral systems are much simpler and less expensive. In this regard, Shahin et al. [97] analyzed the hyperspectral data to select the most relevant wavelengths suitable for detecting green barely kernels. They then developed a multispectral imaging system with a single monochromatic camera and a high-speed filter wheel supported with three different band-pass filter centered at wavelengths of 580, 671 and 730 nm as nominated by the hyperspectral data analysis. The study indicated that the band ratio of 671/730 nm was very effective in scoring the greenness of barely kernels. Through experimentation, a threshold ratio of 671/730 = 75% helped in marking the transition to green kernels that related to the detection level by visual inspection; meaning that kernels having a band ratio less than 75% were considered green. Indeed, this is the first study that designed and developed a fully-functional multispectral imaging system using wavelengths resulting from the analysis of hyperspectral images of seed samples.

In another study, Sumriddetchkajorn et al. [98] used a multispectral imaging system supported by simple image processing techniques for the identification of eight different breeds of rice. Under an ultraviolet excitation light at 265 nm (UVC), only two fluorescent wavelength bands at 540 nm and 575 nm selected by a liquid crystal tunable optical filter were used during image acquisition and simple image thresholding, blob filtering and blob analysis techniques were used for milled rice breed discrimination. The incident UVC light (at 265 nm) shining on milled rice grains was emitted with the corresponding fluorescent radiation and then received through the imaging lens. The tunable optical filter is electrically controlled in such a way that only the desired wavelength spectrum bands can get through to the 2-D digital camera with 45.1 ms switching time between them. The pixel fluorescence intensity at both images were then extracted for all rice breads and used as the basis of breed discrimination. The proposed method showed great potential in distinguishing rice breeds based on their amylose content. Furthermore, due to their action as germicidal light, UVC light used during rice breed identification can also kill unwanted germs attached to milled rice grains.

For manufacturing pasta, the current EU law permits up to 3% (*w*/*w*) adventitious contamination of durum wheat (*T. durum*) with other common wheat (*T. aestivum*), and any malpractice during manufacturing by adding a higher percentage is considered a kind of adulteration that should be instantly prohibited. In a study conducted for seed authentication, Wilkes et al. [99] evaluated the applicability of multispectral imaging for the quantitation of durum wheat grain samples in relation to pasta authenticity by distinguishing between durum wheat *T. durum* and adulterant common wheat *T. aestivum* cultivars based on the spectral signature of each grain species. The performance of the system was evaluated on the basis of its ability to correctly assign the percentage of adulteration level (0, 0.5%, 2%, 3%, 5%, 10% and 100%) of the wheat test samples. Each grain was represented by thousands of pixels that can be scored based on their closeness to the spectral signature of *T. aestivum* or *T. durum*. The normalised-Canonical Discriminant Analysis model (nCDA) algorithm was then applied to calculate the likelihood of each grain being *T. aestivum* or *T. durum* based on composite spectral signature of the pixels within each grain. The resulting classification images, shown in Figure 4, demonstrate close agreement with the assigned percentage of adulteration values. The results of this study indicated the ability of the spectral imaging in testing seed/grain adulteration.

### 4.2. Prediction of Seed Structure

Detecting some key features in seeds such as desiccation represents another challenge in checking the purity of seeds and its acceptability and shelf life because it affects viability and germination ability. In this regard, Jaillais et al. [12] utilized an illumination-based multispectral imaging system supported with eight LEDs operating in the 360–950 nm range (360, 370, 400, 470, 524, 626, 875 and 950 nm) to investigate the effects of desiccation on the texture of wheat endosperm and detecting different desiccation regimes (fast and slow drying). They developed a specific software program to define and control the integration time of the camera and the selection of the LEDs and the photon integration times were adjusted experimentally to prevent any under- or over-exposition. A colour image was acquired at each single LED yielding a total of 3 × 8 = 24 image slices for each single seed. The spectral data extracted from the seeds was explored by PCA to get an overview of the systematic spectral variation in the spectral data and to explore the possibility of grouping the seeds of similar spectral profiles. The results and multispectral image analyses revealed that the change in endosperm texture could be observed directly on the color image obtained with illumination at 360 nm. Under slow-drying conditions, the endosperm texture of seeds remained floury from the start to the end of desiccation treatment; meanwhile, the endosperm texture changed and became glassy under a fast desiccation process.

The same Jaillais et al. research team [73] used the same multispectral imaging system configuration for acquiring reflectance images of both final milled wheat fractions and wheat kernels to identify histological origin of the mills in the kernels. In other words, the main objective of this study was to determine if the spectral characteristics of the milled products can be exploited for relevantly labeling the histological regions in images of wheat kernels. The multispectral images (of 24 sub-image slices each) were processed to create pseudocolour color images to show the proportions of each milled product extracted from the kernel tissues by computing probability images from the multispectral images of kernel sections. The intensities in such images indicate the localization from which a given milled product was extracted. For example, the probability image of coarse bran (CB) shows that this milled product was mainly extracted from the out-layer part of the grain where the intense red color was mainly located at the periphery of the kernel. Thence, the purest semolina (SE) appeared to be extracted from the endosperm region and the break flours (BF) had less clearly identified spectral signature to be extracted from a specific location (Figure 5).

Chevallier et al. [100] employed a similar multispectral imaging system to investigate the potential of such a system to characterize the heterogeneity of food materials (maize, pea, soya bean meal, wheat). Spectral data of the samples were extracted and then analyzed by partial least squares-discriminant analysis (PLS-DA). When the developed model was applied to the original multispectral images, a good visualization was achieved and the proportion of correctly identified pixels observed on the validation set was 75%, 83%, 98%, and 89% for maize, pea, soya bean meal, and wheat, respectively.

A fully mechanized LED-pulsed multispectral system was designed and developed by Pearson et al. [62] using three visible and three near-infrared LEDs with peak emission wavelengths at 470, 527, 624, 850, 940 and 1070 nm for sorting wheat kernels. The multispectral data of wheat kernels were rapidly collected by operating one LED at a time and consequently recording the reflected light from wheat kernels as they dropped off a 20-cm long feeder chute. Sorting of the kernels was conducted by pneumatic system in which an air valve diverted the wheat kernels to the classification assignment. The system achieved high performance in three different applications: (i) separating white and red wheat kernels, (ii) separating undamaged wheat kernels from those ones damaged by *Fusarium* head blight, and (iii) separating wheat kernels based on their average protein content to three classes i.e., high, medium, and low.

By recording sequences of images by changing the spectral conditions, Novales and Bertrand [101] acquired 12 multispectral images at different combinations of excitation and emission filters to identify kernel tissues based on their natural fluorescence characteristics. They used three excitation filters 360, 400 and 510 nm and four emission (fluorescence) filters at 436, 486, 508 and 546 nm by using two filter wheels to select the excitation and emission wavelengths. The most discriminant images among the combined 12 acquired fluorescence images were selected by stepwise discriminant analysis for the identification of kernel tissues. Images were labeled according to the nature of the tissues by attributing each pixel in the sequence of images a qualitative group number corresponding to each tissue. The resulting labeled images exhibit visualized information about spatial distribution of the seed compositional components. As each tissue has a distinct fluorescence behavior, the results revealed that five main tissues (horny endosperm, floury endosperm, germ, coats and tip cap) can be distinguished with more than 98.8 % of the pixels being correctly identified.

### 4.3. Prediction of Seed Viability, Germination Ability and Vigor

In practical applications from farmers’ point of view, seed companies and seed warranty determination, it is very crucial to have some knowledge about seed viability status before sowing to have information for yield prediction [54]. Seed deterioration affects overall germination performance, germination speed and uniformity, seedling emergence and growth, storability, as well as susceptibility to environmental and biological stresses, thereby resulting in a large number of abnormal seedlings and poor plant development. Even though the damaged seeds may have a reasonable germination percentage, such seeds planted under unfavorable conditions rarely produce healthy plants. Thus, knowledge regarding seed vigor and viability is extremely significant for optimizing a future profitable production of the seed. In addition to physical and chemical methods, the ordinary methods used for detecting viability and seed damage involve measurements of seedling growth characteristics as well as physiological and biochemical determination such as conductivity measurement, staining, and enzyme activity determination [15]. The ordinary NIR spectroscopy as well as hyperspectral imaging technology has been used successfully in predicting viability, germination capacity and vigor of different seed cultivars depending on their reference spectral fingerprints [36,41,54,55]. 

In this application, multispectral imaging systems employed in the visible and NIR regions (375–970 nm) with 19 wavelength bands has been tested by Olesen et al. [102] in viability detection of castor (*Ricinus cummunis* L.) seeds. The data of mean intensity for each individual seed were analyzed by nCDA transformation to correlate these data with viability values obtained by the ordinary tetrazolium test for every individual seed. In viability scoring by the tetrazolium test, seeds that developed red colouring of the embryo indicates enzyme activity and were therefore scored as viable seeds and the remaining unstained seeds were considered dead (unviable) seeds. The results revealed that viable seeds were distinguished from dead seeds with accuracy of 92 and 96% in the calibration and validation data sets, respectively. As shown in Figure 6b, the transformed images by nCDA clearly classified the seeds into viable and unviable seeds; resulting in red appearance for the dead yellow seeds and blue appearance for the viable black and grey seeds. The nCDA transformation made the separation very distinct, and a simple threshold was enough to separate stained portions from unstained pixels. This result was confirmed by tetrazolium test performed on cut seeds, in which a similar pattern as for the whole seed was observed (Figure 6d). The yellow seeds display no staining (Figure 6e) whereas the grey and black seeds stain the living tissue red, based on dehydrogenase enzyme activity. The obtained results confirmed the capability of the multispectral imaging technique as a non-destructive method in castor seed viability testing.

Shetty et al. [103] investigated the potential of multispectral imaging in predicting germination capacity and germ length of spinach seeds. Spectral data in the visible and near infrared (395–970 nm) as well as Haralick texture features based on the gray level co-occurrence matrices (GLCMs) were extracted from every single seed in the multispectral images. Spectral data involve the mean, median, minimum, maximum and standard deviation values of reflectance intensities of the seeds. The Haralick texture features included “entropy”, which measures the uniformity of the histogram; “angular”, the second moment measuring the opposite of entropy; “contrast”, which measures the local variation in the image; “correlation”, which measures the linear dependency of gray levels of neighboring pixels; and “inverse difference moment or local homogeneity”, which measures the closeness of the distribution of elements in the GLCM to the matrix diagonal. After germination test, seeds were classifies into four classes (non-germinated seeds, germinated seeds with germ length less than 3 mm, seeds with germ length between 3 and 10 mm, and seeds with germ length larger than 10 mm). Spectral and morphological features were modeled against the germination scores and germ length by partial least squares discriminant analysis (PLS-DA). The results revealed that larger spinach seeds had not only higher germination potential, but also bigger germ length compared with smaller seeds.

### 4.4. Detection of Defects, Infection and Seed Health

Seed health is directly affected by any damage occurred to the seeds as well as the invasion of disease and insect infestation. Seed damage may result from poor harvesting and storage conditions; from rough handling during cleaning and processing or even from disease and/or insect infestations. Such damages may be external, such as cracks in the seed coat or splitting of the seed which may also lead to internal damages and sometimes germinating and vigor losses [104]. Beside losing the germination capacity and producing abnormal seedling growth, infections caused by fungi and viruses usually led to serious problems for seed growers and producers. Therefore, many countries have stipulated strict quarantine rules for importing seeds from zones where specific infections may be endemic [105,106]. Consequently, it is very crucial to detect the infection level of the seeds accurately right after harvest and to optimize methods to reduce the infection on the seeds before sowing. Seed health tests are often performed on representative samples of seed lots, but availability of automatic system to analyze seed lots in individual-seed basis would be advantageous For instance, detection of fungi on seed is performed usually by inspection of the dry seed, washing tests, incubation methods, embryo count method or seedling symptom tests, but employing a rapid, non-destructive method would help in saving time and cost spent compared to the ordinary evaluation methods. Although the International Seed Testing Association (ISTA) has developed accredited protocols for seed health testing of many species, there is no accredited protocol for seed health testing for all species [6]. Thus, development of a high-speed system based on optical methods having the potential to rapidly detect and physically remove seeds severely diseased or contaminated by fungi, or infested internally by insect larvae or pupae is very crucial [107]. In this regard, multispectral imaging has found its way towards automating the inspection and evaluation practices of seed health and proved to be useful in the separation of infected seeds from uninfected seeds in several species [74,78,81].

As seed health tests are time consuming processes and require substantial training for characterization of pathogenic fungi on seed, multispectral imaging system (395–970 nm) was used by Olesen et al. [78] for identifying surface properties of different fungal infections in spinach seeds and for discrimination among healthy and seeds infected seeds with different *Fusarium* strains (*Verticillium* spp., *Fusarium* spp., *Stemphylium botryosum*, *Cladosporium* spp. and *Alternaria alternate*). They used canonical discriminant analysis (CDA) for separation of image pixels based on their mean intensity and Jefferies-Matusita (JM) distance for classification and modeling spectral data which gave a separation accuracy of 26-88% between uninfected and *Fusarium* spp. infected seeds.

Vrešak et al. [81] utilized a multispectral imaging system (375–970 nm) with 19 wavelength bands to discriminate 27 winter wheat (*Triticum aestivum* L.) varieties and nine triticale (*Triticosecale* Wittm. & Camus) varieties in terms of their resistance to fungal infection. The same method outlined by Olesen et al. [78] was followed in which the nCDA method for multispectral images analysis was used to perform the data transformation on all 19 bands. In this method, some regions of interest (ROIs) representing the infected and uninfected (healthy) tissues of the seeds based on the visual signs of infection or unification status on dry seeds were marked and their reference spectral patterns were used in training the model during data transformation. The CDA transformation was used to minimize the distance between observations within the group and maximize distance between the known groups (36 different varieties or/and infected and uninfected parts of the seeds). The supervised transformation by nCDA was used to divide the images into regions of interest. To determine the accuracy of detection, the nCDA transformation was then applied on all images and examined whether the pixel classification results corresponded to the manual visualization. Classification results shown in Figure 7 illustrate how accurate the method was in identifying infection groups in which the parts infected with *Alternania* sp. appeared in orange and parts infected with *Fusarium* sp. appeared as light blue; meanwhile, the uninfected healthy parts appeared in dark blue.

In another study for detecting different defects in maize kernels, Sendin et al. [43] used a multispectral imaging system in the UV, visible and NIR regions (375-970 nm) with 19 discrete wavelengths. They used principle component analysis (PCA) to explore spectral variation among samples and to visualize the mean difference between sound and defected seeds. The defects include damages from heat or water, fungal damage from *Fusarium* or *Diplodia*; pinked white or yellow maize, broken kernels; and foreign matters (wheat, soy, sunflower seeds, sorghum and maize plant material). Moreover, partial least squares discriminant analysis (PLS-DA) models were used for distinguishing between sound maize and undesirable materials. The PLS-DA model had good classification accuracies ranging from 83% to 100%. 

The study also concluded that wavelengths related to red, pink and blue carotenoid and anthocyanin pigments (505, 525, 570 and 590 nm), and NIR wavelengths 890, 940 nm (associated with fat) and 970 nm (associated with water) are considered as relevant wavelengths for discrimination. Figure 8a shows the PCA score image to demonstrate how the heat damaged seeds are entirely different from the sound seeds. By using supervised discrimination method by PLS-DA, the discrimination among heat damage and sound maize kernels was very obvious and easier to observe (Figure 8b). This result reveals high potential to industrial applications due to shorter time used in image acquisition and analysis as well as lower cost compared with hyperspectral imaging systems. 

For the detection of fungal contamination, Jaillais et al. [74] designed and developed a non-destructive methodology based on LED-illuminated multispectral imaging and chemometric to detect *Fusarium* head blight infection in wheat kernels. For each sample, eight RGB images with a spatial resolution of 524 × 702 pixels each were acquired and merged into an “image-cube” dimensioned 524 × 702 × 24. Spectral data were extracted from every single kernel and then analyzed via PCA and stepwise multilinear regression (SMLR) methods. The developed SMLR model was then applied to estimate the degree of contamination of each pixel in the image to produce ’predicted’ classification images with high values indicating a high local contamination. The PCA results of the data indicated the possibility of separating the ‘susceptible’ and ‘resistant’ genotypes of wheat kernels (Figure 9a,b). Similarly, SMLR models applied on multivariate images provide ‘predicted’ classification images of the wheat kernels. In such classification image, the vales of the pixels are coded based on their predicted degree of contamination. Based on the visualization of contamination in this image, a kernel with high contamination level exhibited more red pixels indicating that the ‘susceptible’ kernels had a majority of red pixels while ‘resistant’ ones exhibited more blue pixels (Figure 9c,d).

Because mycotoxins, the toxic metabolites of certain filamentous fungi, cause very severe health problems and are considered as potent carcinogens causing many human deaths per annum, Kalkan et al. [108] applied multispectral imaging for detecting aflatoxin-contaminated hazelnut kernels and chili pepper. The system was illuminated by UV-A light having a peak intensity at 365 nm and supported with a filter wheel for holding 12 different band-pass filters with 10 nm full width half maximum (FWHM) in the range of 400–510 nm and two other filters at 550 and 600 nm with 70 and 40 nm FWHM, respectively. By using an algorithm of local discriminant bases (LDB) to obtain localized information for spectral signals and image classification, the detection accuracy of aflatoxin-contaminated red peppers and hazelnut kernels was 92.3 and 79.17%, respectively. More precise overall classification accuracy of 95.67% was achieved in classifying hazelnuts as fungal infested kernels without considering their aflatoxin concentrations.

### 4.5. Detection of Pest Infestation

Pests inflict their damage on seeds mainly by feeding on the endosperm leading to sever losses in both seed weight and quality. When pest species feeds on the germ, it results in poor seed germination and less seed viability [109]. In addition to the direct consumption of different seed portions, insect pests contaminate the infested seeds with their by-products such as excretion, molting, dead bodies and their exuviae as well as encouraging the infection with bacterial and fungal diseases, which is not commercially desirable. Symptoms of infestation include the presence of insect themselves, their by-products, emergence holes made by insects in the seeds, raising temperature and humidity inside the packages, and powdery residues below and surrounding the seeds. 

Ma et al. [110] tested a multispectral imaging system in the visible and NIR ranges (405–970 nm) for discrimination of insect-infested, moldy, heterochromatic, and rancidity in sunflower seeds nondestructively. Compared with intact sunflower seeds, the infested sunflower seeds showed higher spectral reflectance in the range of 780–970 nm due to severe changes in the physicochemical properties occurred in the infested seeds. By using Fisher’s stepwise linear discriminant analysis based on 10 feature wavelengths (435, 470, 525, 630, 645, 660, 700, 850, 910, and 970 nm) out of the original 19 wavelengths, authors achieved an excellent classification accuracy of more than 97% for detecting intact sunflower seeds. They also indicated that intact sunflower seeds with different degrees of rancidity could be precisely clustered using principal component analysis-cluster analysis (PCA-CA).

## 5. Real-Time Systems for High-Throughput Applications

The main challenge of seed industry in the recent century is to acquire consistent information about overall seed quality and safety during the production, processing, handling, marketing and distribution chains, to satisfy consumer demands and the requirement for market expansion. During the last couple of decades, the advancement in image acquisition protocols along with rapid image processing algorithms has permitted quantifications of many morphological and physico-chemical features of the inspected seeds. Conventional methods for measuring seed quality traits by visual inspection of seed intactness and checking germination capacity, seed health and freedom from disease infestation are criticized as being costly and tedious analytical procedures. In this context, imaging techniques, especially spectral imaging methods, have been proved to be efficient techniques for rapid monitoring of numerous seed quality traits. Thence, it was very critical for seed community to explore the potential of this technique in real-time implementations [86]. Because hyperspectral cameras are expensive and not suited for high-throughput testing, the development of more simple and affordable devices is necessary [74]. Actually, most of the research works in using hyperspectral imaging system were basically conducted in the hope of developing a multispectral imaging system that could be implemented in real-time applications for predicting and visualizing various intrinsic or morphological features. Many published articles claimed to have already developed multispectral imaging systems for prediction and visualization of certain quality attributes but very few of these works were crowned with a robust design that could be installed in on-line scenarios. Such research attempts focused on scanning seed samples in laboratory-based hyperspectral imaging systems followed by selecting some key wavelengths most related to certain quality features without using such wavelengths in building an independent and fully-functional multispectral imaging system. Although the process of selecting the ideal wavelengths using various machine learning routines is substantially important, the main challenge is to utilize such wavelengths in establishing a multispectral machine vision system. Despite this wide range of usages of multispectral imaging systems, few systems are commercially available in the market and most of them are complicated, expensive and bulky [60]. Thus, it is extremely important to develop a computer-integrated system that has the ability to deliver a large amount of seeds to the imaging system (e.g., through a conveying system) and subsequently sort the seeds into different fractions using robotic picking or some kind of pneumatic or mechanical gate system [77]. The timely detection of seed quality attributes that do not meet quality specification standards is advantageous, as it allows immediate rejection of the imported lots, thereby saving time and money. Furthermore, seed quality evaluation directly after production and drying process is necessary, because it allows continuous control of the major quality parameters. In addition, timely determination of seed viability, intactness and wholesomeness is required for process control, as the occurrence of defect-causing features such as fungal infection or disease infestation are strongly dependent on the seed health conditions.

In general, the real-time or near real-time assessment of quality attributes of seeds and/or any relevant bio-product in harsh environment requires a better understanding of the critical quality attributes of the seeds being analyzed. Also, a balance between the cost of equipment including hardware and software, the detection speed of the imaging system, the detection accuracy of the model, and the service life of the equipment must be all considered during building a comprehensive real-time spectral imaging system to move the operations the from the laboratory setting to online systems in the industry [86]. Because the development of an automated system for real-time evaluation of seed quality during all stages of the production chains in very essential, this system should be readily available to the industry and easy-to-use without requiring special expertise from end-users. In addition, the system must be accurate and reliable for providing rapid, non-destructive, low cost analysis with minimum or no sample preparation and have the potential to analyze multiple attributes simultaneously [40]. Although the developed system may be automated, their implementation on seed applications could be not straightforward if the relevant wavelengths have not been carefully selected or when the calibration procedure has not been precisely conducted [18]. Consequently, the procedure of selecting the ‘*optimal*’ wavelength is considered the most important step in developing and establishing an a high-throughput system that has the capability of performing the inspection and quality control decision in real time [111].

Indeed, the real-time operations are very complicated processes that require acquisition of images at several bands and process such images simultaneously. The procedure of selecting the ideal wavelengths can either be performed before the modeling or even during the modeling process by discarding the unnecessary wavelengths that add complexity of the model and reduce its efficiency in classification or prediction. These unwanted wavelengths make model interpretation difficult and affect the classification and prediction efficiency of the model [41]. There are several methods for selecting such wavelengths [91]; but there is no only one magic method that fits all applications. The method of wavelength selection should also identify the optical center frequency and bandwidth (FWHM) that provide optimal discrimination in the multispectral imaging system [108]. By using the optimal wavelengths and the appropriate modeling algorithm, the system will allow the visualization of various attributes within the sample because regions with similar spectral patterns tend to appear in the same colour which facilitates its separation and categorization by the system. Moreover, using fewer wavebands in multispectral imaging designs can lead to faster sensor systems, thus reducing total integration time and increasing the number of samples inspected per second [41].

## 6. Conclusions

The potential of spectral imaging technique in providing rich and valuable information about seed quality traits and phenotyping parameters is powered from the robust integration among spatial imaging, spectroscopy and chemometrics tools that makes this technique an ideal tool in studying various morphological, physicochemical and physiological properties of seeds. This extraordinary capability has enticed researchers to exert cooperative efforts in developing fast, accurate and low-cost spectral systems to be installed in seed and grain industry. With the advantage of acquiring three-dimensional data across a wide range of the electromagnetic spectrum, state-of-the-art multispectral imaging along with relevant multivariate chemometric analysis has been successfully implemented, not only for food quality and safety control purposes but also in dealing with critical research challenges in seed science and technology. This paper provided an overview of the previous research activities employed for quality evaluation and safety analysis of different seed cultivars using multispectral imaging techniques. The structure of all possible designs and image acquisitions modes of these systems were reviewed and the overall merits and demerits of such configurations for specific usability and applicability were also discussed. This review continues from where previous reviews stopped and focuses only on real implementations of fully-operated multispectral imaging systems for quality assessment of different sorts of seeds. Thence, this review comprehensively reviewed research attempts only devoted to fully-operated multispectral imaging systems and does not consider those ones that just dedicated to selections of key wavelengths from hyperspectral data analyses without building up independent multispectral imaging systems. This makes this review article the first attempt in summarizing all published papers in multispectral imaging applications for seed phenotyping and quality monitoring. The review has emphasized the capability of the multispectral imaging with different chemometrics algorithms in characterizing physicochemical quality traits, predicting physiological parameters, variety identification and classification, detection of damages, defect, pest infestation and seed health. By careful consideration of all limitations and challenges faced by this technology, it is anticipated that the multispectral imaging technique can be moved from laboratories to practical applications in the form of real-time seed monitoring systems that meet the requirements of the modern industrial control and sorting systems. The decreasing cost and increasing speed and capability of computer hardware and artificial intelligence will push this mission forward and make this technology more attractive for prospective usages in quality control and automatic inspection of seeds.

## Figures and Tables

**Figure 1 sensors-19-01090-f001:**
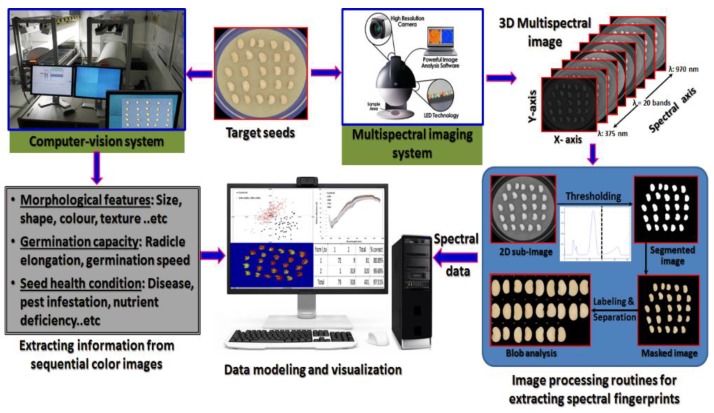
Schematic representation of a computer-aided image analysis system for seed quality evaluation based on computer-vision and multispectral imaging techniques.

**Figure 2 sensors-19-01090-f002:**
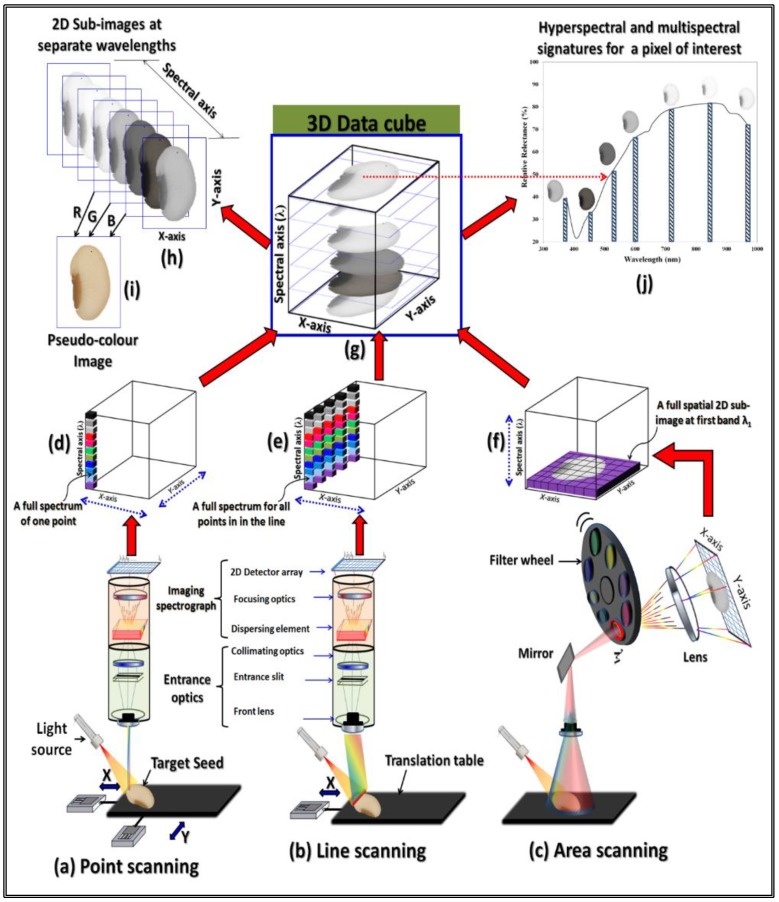
Schematic representation of three approaches used for constructing spectral image cubes. The blue dashed arrows indicate scanning directions in each approach for sequential acquisitions to complete the volume of spatial and spectral 3D data cube.

**Figure 3 sensors-19-01090-f003:**
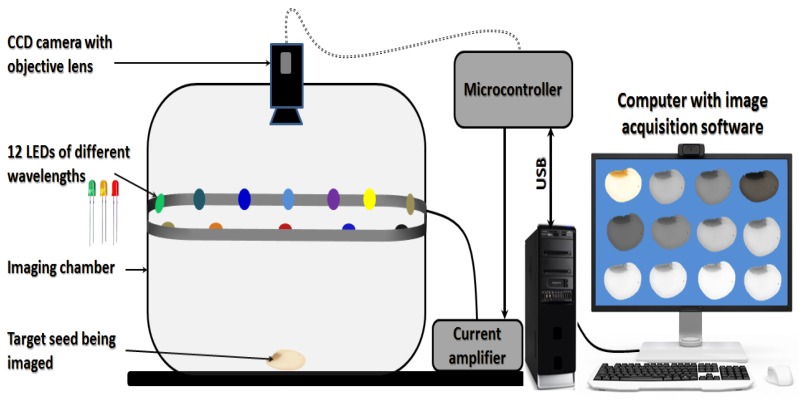
An illumination-based multispectral imaging system supported with an array of 12 LEDs with different nominal emission wavelengths. Each LED is sequentially switched on and synchronized with the camera trigger for image acquisition using a pre-programmed microcontroller.

**Figure 4 sensors-19-01090-f004:**
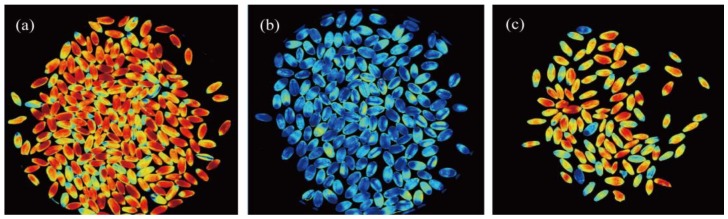
Images generated using nCDA algorithm for: (**a**) 100% *T. durum* wheat grains; (**b**) 100% *T. aestivum* wheat grains; (**c**) 10% adulteration of *T. durum* wheat grains with *T. aestivum* wheat grains [99].

**Figure 5 sensors-19-01090-f005:**
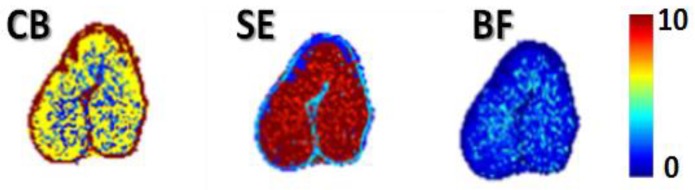
Probability images (represented in pseudo-colors) associated with a multispectral image of wheat kernel. High probability percentage indicates the region of the grain from which a given milled product was extracted. CB: coarse bran, SE: pure semolina and BF: break flours (Modified from Jaillais et al. [73]).

**Figure 6 sensors-19-01090-f006:**
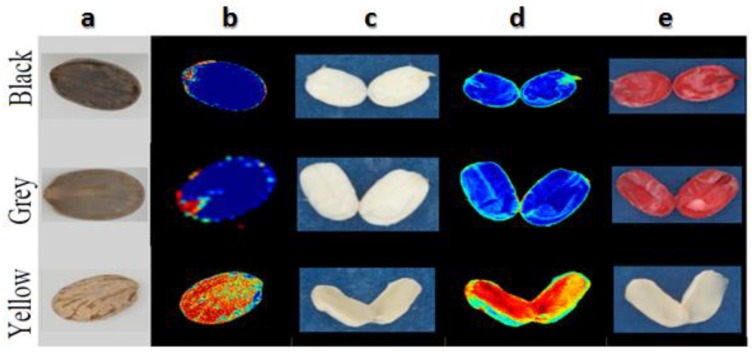
Classification of castor seeds into three classes based on visual colour of seed coat (yellow, grey and black). (**a**) RGB images of the intact seeds; (**b**) transformed images of intact seeds by nCDA; (**c**) RGB images of cut seeds; (**d**) transformed images of cut seeds by nCDA and (**e**) RGB images of cut seeds after immersion in tetrazolium solution (adapted from Olesen et al. [102]).

**Figure 7 sensors-19-01090-f007:**
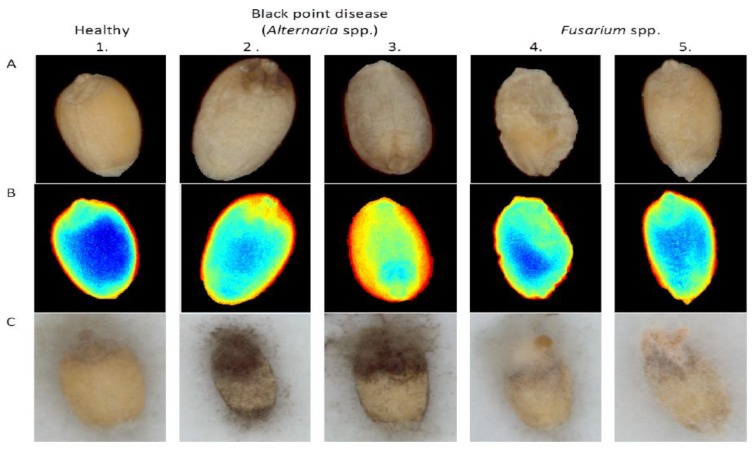
Difference between healthy and infected seeds by black point disease (*Alternaria* sp.) and *Fusarium* sp. (**A**) RGB-captured images, (**B**) nCDA-transformed images, and (**C**) RGB images after seed incubation (adapted from Vrešak et al. [81]).

**Figure 8 sensors-19-01090-f008:**
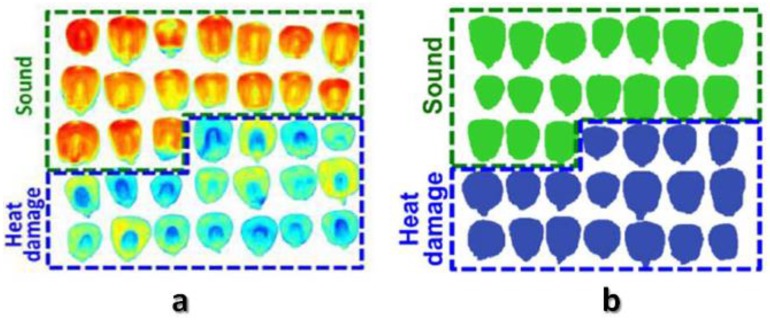
Defect detection on maize kernels (**a**) PCA scores image showing the clear separation between heat damage class and sound class maize, (**b**) classification image resulting form PLS-DA for discrimination between heat damage class and sound class maize kernels (adapted from Sendin et al. [43]).

**Figure 9 sensors-19-01090-f009:**
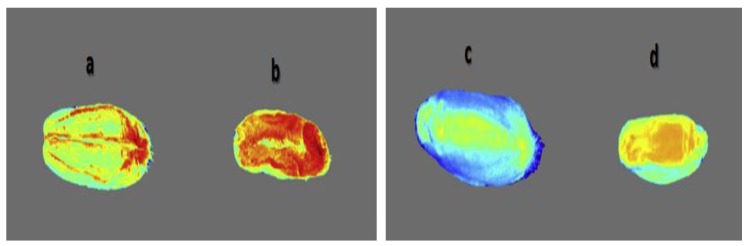
PCA score-images issued from PCA on whole images for the (**a**) resistant and (**b**) susceptible wheat genotypes; and the predicted contamination images resulting from SMLR model in the (**c**) resistant and (**d**) susceptible wheat genotypes kernels in which a kernel highly contaminated have a lot of red pixels (adapted from Jaillais et al. [74]).

**Table 1 sensors-19-01090-t001:** Recent applications of multispectral imaging systems in phenotyping and quality evaluation of various seeds.

Seed Type	Applications (Attributes)	Spectral Range	No. of WL	Data Modeling	References
Spinach	Seed health detection	395–970	19	CDA	[78]
	Prediction of germination & germ length	395–970	19	PLSDA	[35]
Tomato	Variety identification	375–970	19	PCA & nCDA	[94]
	Variety identification	375–970	19	nCDA	[95]
	Cultivar discrimination	375–970	19	PLSDA & SVM-DA	[79]
Wheat	Characterization of desiccation	360–950	8 × 3	PCA	[12]
	Seed health & variety separation	375–970	19	nCDA & k-NN	[81]
	Seed authentication	375–970	19	nCDA	[99]
	Fungal contamination detection	360–950	8 × 3	PCA & MLR	[74]
	Identification of the histological origin	360–950	8 × 3	PCA & LDA	[73]
	Variety discrimination, fungal detection & prediction of protein content	470–1070	6	LDA	[62]
Rice	Discrimination between transgenic & non-transgenic seeds	405–970	19	PLSDA, LS-SVM & PCA-BPNN	[76]
	Variety discrimination	365–970	46	k-NN & nCDA	[65]
	Breed identification	540 & 575	2	Blob analysis	[98]
Maize	Histologic labeling of seed parts	436–546	12	LDA	[101]
	Discrimination between haploid and diploid kernels	375–970	19	nCDA	[77]
	Defect detection	375–970	19	PCA & PLSDA	[43]
Triticale	Seed health & variety separation	375–970	19	nCDA & k-NN	[81]
Sunflower	Detection of mold and insect infestation	405–970	19	LDA & PCA-CA	[110]
Castor	Viability prediction	375–970	19	nCDA	[102]
Hazelnut	Detection of aflatoxin contamination	400–600	14	LDB	[108]
Chili pepper	Detection of aflatoxin contamination	400–600	14	LDB	[108]
Soybean	Discrimination between conventional and transgenic seeds	405–970	19	PLSDA, LS-SVM & PCA-BPNN	[96]
Pea	Discrimination among maize, pea, soybean & wheat	360–950	8 × 3	PLS-DA	[100]
Barely	Greening detection	580–730	3	Band ratio	[97]

WL: Wavelength; CDA: normalized canonical analysis; nCDA: normalized canonical analysis; PCA: Principle component analysis; PLS: partial least squares; PLS-DA: partial least squares-discriminant analysis; SVM: Support vector machine; SVM-DA: Support vector machine-discriminant analysis; k-NN: k-nearest neighbor algorithm; LDA: linear discriminant analysis; LS-SVM: Least squares support vector machine; BPNN: Back-propagation neural network; MLR: multilinear regression; LDB: local discriminant bases (LDB) algorithme; PCA-CA: Principle component analysis-cluster analysis.

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
