# Peer review of "Recent Applications of Multispectral Imaging in Seed Phenotyping and Quality Monitoring—An Overview"

_sensors, 2019, doi:10.3390/s19051090_

Round 1
Reviewer 1 Report
This manuscript is a review about the use of multispectral imaging and advanced machine learning methods in seed phenotype and quality monitoring. The topic is ok, but the content does not exactly correspond to the title. Almost half of the pages only propose the concept of multispectral imaging, some of which are not necessary actually. The rest part involve the applications of multispectral imaging for seed quality monitoring while the point of advanced machine learning methods does not be highlighted as the title given. The structure of the article should be adjusted to complement the algorithms and applications of advanced machine learning, not just the traditional ones.
Author Response
Authors:
Thank you indeed for the time devoted in reading and commenting on our manuscript and for the critical point you highlighted in your revision. As this is the first review article in the applications of fully-operated multispectral imaging systems for seeds phenotyping and quality monitoring, we had to pay the reader’s attention to the principles and fundamental of different multispectral imaging techniques and the most usable methodologies in implementing this technique for this particular task. We intentionally avoid to mention all details about all machine learning methods (in terms of mathematical background, equations, formulas, theories, hypothesis..etc as the reader can find all of this information easily in any relevant text book) because the main purpose was to go directly to the real applications in seed phenotyping. Based on that, we suggest changing the title of this review article as follow:
Old title: Multispectral imaging and advanced machine learning methods for seed phenotyping and quality monitoring – an overview
New title: Applications of multispectral imaging systems for seed phenotyping and quality monitoring – an overview
As the new title implies, the article will be completely devoted to the multispectral imaging and its direct applications in the seeds phenotyping without stating details about machine learning methods.
Reviewer 2 Report
The work carried out for this manuscript is valuable and has merits for publication in Sensors. In my opinion it requires a minor revision before be suitable for publication, addressing the following aspects:
Please, change to cursive the species names (Fusarium,etc).
Multispectral imaging has also been evaluated for nondestructive determinations of chemical composition (protein, lipid, moisture, pigments, caffein,..etc) of seeds (peanut,rice,sunflower seed, wheat,maize, coffee bean, etc). I suggest to include a Table for introducingrecent progresses of multispectral imaging with appropriate chemometrics (spectral range, optimal model, accuracy, ... ) in theseed quality monitoring.
Author Response
Comment: The work carried out for this manuscript is valuable and has merits for publication in Sensors. In my opinion it requires a minor revision before be suitable for publication, addressing the following aspects:
Authors: Thank you indeed for your time and effort you made and for the positive opinion about our manuscript and also for the constructive comments.
Comment:Please, change to cursive the species names (Fusarium,etc).
Authors: Changes made throughout the whole manuscript
Comment: Multispectral imaging has also been evaluated for nondestructive determinations of chemical composition (protein, lipid, moisture, pigments, caffein,..etc) of seeds (peanut,rice,sunflower seed, wheat,maize, coffee bean, etc). I suggest to include a Table for introducing recent progresses of multispectral imaging with appropriate chemometrics (spectral range, optimal model, accuracy, ... ) in the seed quality monitoring.
This review comprehensively highlights all research attempts devoted to only fully-operated multispectral imaging systems and did not consider those ones that just utilized some key wavelengths extracted from hyperspectral data analyses without building independent multispectral imaging systems. Based on this statement mentioned even on the abstract, we only reviewed research works conducted on fully-operated multispectral imaging systems for different applications in seed quality assessment. In this regard, only one article was found in using multispectral imaging systems for the prediction of protein content in wheat seeds [62]. All details about spectral range, number of wavelengths and the optimal data modeling were mentioned in Table 1 of the manuscript.
Reviewer 3 Report
The manuscript provides a very thorough description of the topic and I found it particularly informative. I think this will be a valuable reference for the field.
The manuscript was rather long, but as long as the journal doesn't have a word/page limit, this isn't an issue.
Author Response
The manuscript provides a very thorough description of the topic and I found it particularly informative. I think this will be a valuable reference for the field.
The manuscript was rather long, but as long as the journal doesn't have a word/page limit, this isn't an issue.
Authors: Thank you so much for this positive opinion about our literature review paper. We appreciate your valuable time you devoted in reading the article.
Round 2
Reviewer 1 Report
There are several figures in the manuscript have already been published elsewhere. Authors should obtain permission from the copyright owners and include evidence that such permission has been granted when submitting their papers.